# Comparison of Manual versus Semi-Automatic Segmentations of the Stenotic Carotid Artery Bifurcation

Benjamin Csippa [1,*,†], Zsuzsanna Mihály [2,†], Zsófia Czinege [2], Márton Bence Németh [1], Gábor Halász [1], György Paál [1] and Péter Sótonyi, Jr. [2]

[1] Department of Hydrodynamic Systems, Budapest University of Technology and Economics, Műegyetem rkp. 3., 1111 Budapest, Hungary; mnemeth@hds.bme.hu (M.B.N.); ghalasz@hds.bme.hu (G.H.); paal@hds.bme.hu (G.P.)

[2] Department of Vascular and Endovascular Surgery, Semmelweis University Budapest, Városmajor Str. 68, 1122 Budapest, Hungary; mihaly.zsuzsanna@med.semmelweis-univ.hu (Z.M.); czinege.zsofia@gmail.com (Z.C.); sotonyi.peter1@med.semmelweis-univ.hu (P.S.J.)

[*] Correspondence: bcsippa@hds.bme.hu

[†] The author contributes the same.

**Abstract:** Background: The image reconstruction of stenotic carotid bifurcation can be managed by medical practitioners and non-medical investigators with semi-automatic or manual segmentation. The outcome of blood flow simulations may vary because of a single mean voxel difference along the examined section, possibly more in the stenotic lesions, which can lead to conflicting results regarding other research findings. The aim of our project is computational geometry reconstruction for blood flow simulations to make it suitable for comparison with plaque image analysis performed by commercially available software. In this paper, a comparison is made between the manual and semi-automatic segmentations performed by non-medical and medical investigators, respectively. Methods: 30 patients were classified into three homogeneous groups. Our group classification was based on the following parameters: plaque calcification score, thickness, extent, remodeling and plaque localization. The images in the first group were segmented individually by medical practitioners and experienced non-medical investigators, the second group was segmented collectively, and the last group was segmented individually again. Cross-sections along the centerline were extracted, then geometrical and statistical analyses were performed. Exploratory flow simulations were carried out on two patients to showcase the effect of geometrical differences on the hemodynamic flow field. Results: The largest centerline-averaged voxel difference between the medical and non-medical investigators occurred in the first group with a positive difference of 1.16 voxels. In the second and third groups, the average voxel difference decreased to 0.65 and 0.75, respectively. The example case from the first group showed that the difference in maximum wall shear stress in the middle of the stenosis is 30% with an average voxel difference of 1.73. Meanwhile, it can decrease to 4% when the average voxel difference is 0.64 for the example case from the third group. Conclusions: A collective review of the medical images should preceded the manual segmentations before applying them in computational simulations in order to ensure a proper comparison with plaque image analysis. Especially complex pathology such as calcifications should be segmented under medical supervision or after specific training. Non-significant differences in the segmentation can lead to significant differences in the computed flow field.

**Keywords:** three-dimensional imaging; computer-assisted image processing; carotid artery stenosis; computed tomography angiography

## 1. Introduction

The current European and US guidelines for stroke prevention in patients with atherosclerotic carotid plaques are based on the quantification of the stenosis degree. Although recent guidelines reflect that the risk of stroke is related to carotid plaques, it



cannot be attributed exclusively to the degree of stenosis; plaque morphology (geometry and tissue composition) also plays an important role in risk evaluation [1,2]. Therefore, the tradition of using the degree of luminal stenosis as the only imaging marker and indicator of clinical decision-making is challenged by new evidence. Non-invasive identification of carotid artery plaque characteristics may allow risk stratification, predict early revascularization, and may help determine the efficacy of pharmacologic treatment by measuring plaque progression/regression, especially in the case of aggressive pharmacologic plaque management. However, there are no recommendations or exact hints regarding the definition, quantification and qualification determining the role of plaque morphology in stroke risk estimation or at the choice of therapeutic approaches.

### 1.1. Imaging of Carotid Artery Pathology and Its Clinical Relevance

According to Class I, level A recommendation CT angiography (CTA) is suitable for extent and severity evaluation of extracranial carotid stenosis [1]. The advantages of CTA are wide accessibility in daily clinical routine, high spatial resolution, allowing delineation of arterial outlines, well-established imaging protocols, shorter scan times, ensuring reduced motion artefacts [3] and the capacity to accurately define plaque volume and subcomponents, especially when coupled with post-processing analysis techniques [4]. Furthermore, CTA-derived diameter stenosis measurements correlate well with catheter angiographic measurements [5]. The importance and efficacy of semi-automatic segmentation with CTA has been proven to be fundamental in the daily clinical routine in stenosis grading. Four advanced semi-automatic vessel analysis applications were tested in comparison by two expert neuroradiologists. They have a low-to-good sensitivity, good specificity and overall diagnostic accuracy, as well as high interobserver reproducibility in stenosis grading. It was, however, remarked that due to the false-negative classifications, which were caused by ulcerative plaques and observer variation in stenosis and reference measurements, automatic measurements should be checked by an experienced radiologist [6]. Moreover, CTA can be used to assess the volume of atherosclerotic plaques and detect ulcerations. Furthermore, CTA image analysis provides enough detail for morphological analysis and for calcium identification. However, it is difficult to reliably differentiate between soft plaque components because of an overlap in Hounsfield units (HU), as it is not an ideal approach to identify the fibrous cap and it overestimates the stenosis grade because of calcium deposits [7].

### 1.2. Computational Geometries for Blood Flow Simulation in Stenotic Carotid Artery

Computational fluid dynamics (CFD) studies emphasize that elevated wall shear stress (WSS) is frequently associated with high-risk plaque features in coronary arteries [8–10]. Its role as a trigger for rapid plaque progression and expansive remodeling has not been clearly proven. There still remains an ongoing debate whether low or oscillatory shear parameters might be a better indirect indicator for plaque vulnerability [11]. Complex geometries with several anatomical variations [12], such as the carotid bifurcation, affect the hemodynamics to an even larger extent [13,14] than a single vessel malformation and complexity can further increase in the presence of stenoses at regions of a bifurcation. Furthermore, the geometry of the carotid artery differs from individual to individual, leading to an ensemble of different flow features. The results of CARE II study based on multicontrast magnetic resonance (MR) vessel wall imaging suggests that there is a relationship between high wall shear stress and vulnerable plaque features in smaller luminal expansion at carotid bifurcation [15]. This new finding underlines the results of previous articles [9,16], which were criticized before [8]. In a recent article [17], the authors concluded that one parameter alone is not appropriate as a surrogate marker for cardiovascular risk stratification, rather the different markers should be investigated as individual risk factors. Helical flow-based geometric descriptors [17,18] are considered to be good indicators of disturbed flow regions (low or oscillatory WSS regions). However, as pointed out in these review articles [11,15], further investigations are needed to understand the underlying complex mechanisms.

The aim of our study was to reconstruct computational geometries for blood flow simulations from CTA images that are appropriate for comparison with a plaque image analysis using commercially available software. In this paper, a comparison is made between the manual and semi-automatic segmentations performed by non-medical and medical investigators in homogenous (based on main plaque morphology features) patient groups.

## 2. Materials and Methods

Our multidisciplinary working group intended to perform this study according to the plan is summarized in the flowchart of Figure 1.

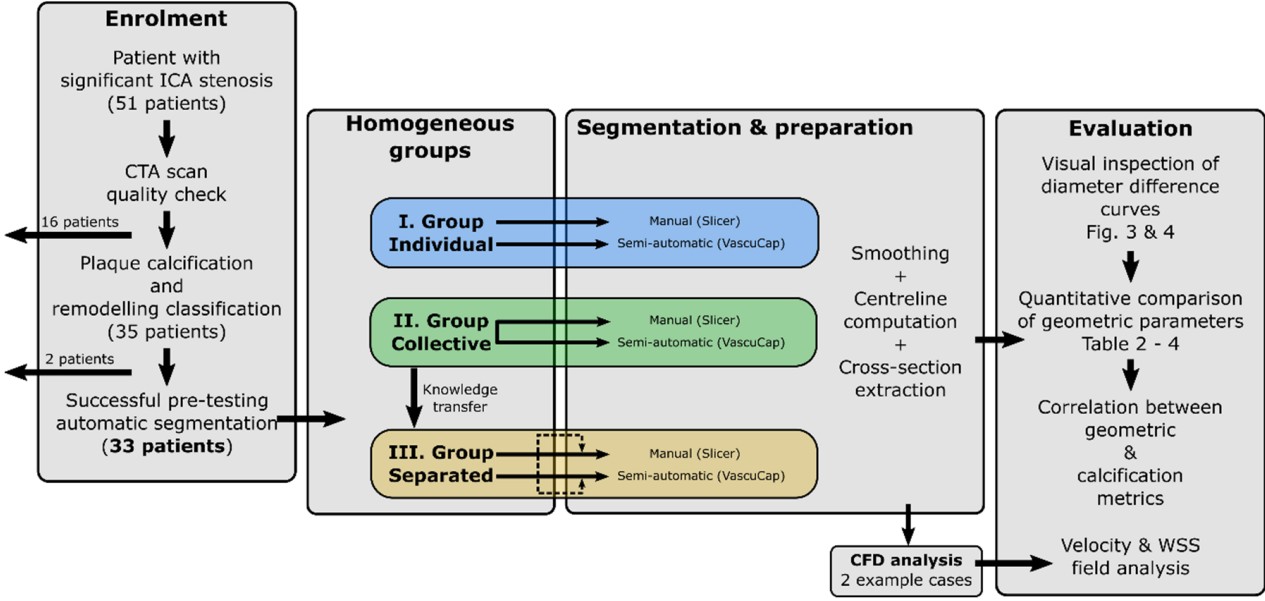

**Figure 1.** Summary flowchart of the main methodological steps: Patient enrolment including patients excluded from the study, homogeneous grouping, segmentations, and subsequent preparations of the three groups with two methods, CFD analysis of two cases and evaluation of the data. (ICA: Internal carotid artery CFD: Computational fluid dynamics WSS: wall shear stress.

### 2.1. Enrolment of Patients

In our prospective study group, 51 patients with significant carotid artery stenosis were enrolled in the Dept. of Vascular and Endovascular Surgery of Semmelweis University between 1 January 2019 and 28 February 2020 (Reg No: NCT03840265 on clinicaltrials.gov). Demographics and comorbidities were recorded. Patients underwent CT angiography as part of standard-of-care diagnostic evaluation using a routine clinical imaging protocol. After the patient enrolment, the CTA image quality was classified in the Likert scale (0–4) for CTA segmentation. The patients lacking the highest quality images (Likert scale less than 4) were excluded, since a high arterial contrast enhancement is advantageous for accuracy [19].

The 35 included CTA images were classified regarding the localization of the stenosis (bifurcation, suprabulbar), plaque calcification (extent and thickness) and plaque remodeling (by ZM and ZC) in consensus reading. The calcification scoring was measured based on Babiarz's study [20]. The plaque remodeling was measured with Yosida's method [21]. The automatic centerline identification was pre-tested, the unsuccessful two cases were excluded. The included images were sorted into three homogenous groups based on the calcification score and plaque remodeling status. Homogeneity was verified by paired Chi-square tests performed between each group pairings (1–2, 2–3, 1–3). Table 1 shows the results of the image evaluation of each patient, sorted by their associated homogenous groups.

**Table 1.** The results of the variables from the image evaluation of the homogenous groups. The following grouping variables were used: plaque localization (1: bifurcation, 2: suprabulbar), plaque calcification (extent 0–4 and thickness 0–4 score is the sum of the extent and thickness values; (Babiarz 2003) [20]) and plaque remodeling (no: 0 yes: 1; (Yosida 2015) [21]).

| Group I.—Individual | | | | | | Group II.—Collective | | | | | | Group III.—Separated | | | | | |
|---|---|---|---|---|---|---|---|---|---|---|---|---|---|---|---|---|---|
| ID | Plaque loc. | Calc. Extent | Calc. Thickness | Calc. Score | Pos. Rem. | ID | Plaque Loc. | Calc. Extent | Calc. Thickness | Calc. Score | Pos. Rem. | ID | Plaque Loc. | Calc. Extent | Calc. Thickness | Calc. Score | Pos. Rem. |
| 005 | 1 | 2 | 2 | 4 | 1 | 003 | 2 | 2 | 2 | 4 | 0 | 002 | 2 | 2 | 2 | 4 | 1 |
| 010 | 2 | 0 | 0 | 0 | 1 | 004 | 1 | 1 | 1 | 2 | 1 | 006 | 1 | 1 | 1 | 2 | 0 |
| 013 | 1 | 1 | 1 | 2 | 1 | 008 | 1 | 1 | 1 | 2 | 0 | 025 | 1 | 1 | 2 | 3 | 1 |
| 021 | 1 | 1 | 1 | 2 | 0 | 030 | 2 | 1 | 2 | 3 | 1 | 027 | 1 | 2 | 2 | 4 | 0 |
| 045 | 2 | 1 | 1 | 2 | 0 | 031 | 2 | 0 | 0 | 0 | 0 | 029 | 1 | 4 | 4 | 8 | 1 |
| 050 | 1 | 0 | 1 | 1 | 0 | 034 | 1 | 2 | 4 | 6 | 0 | 035 | 1 | 1 | 1 | 2 | 1 |
| 051 | 1 | 4 | 3 | 7 | 1 | 040 | 1 | 3 | 2 | 5 | 1 | 036 | 1 | 0 | 0 | 0 | 1 |
| 052 | 1 | 4 | 4 | 8 | 1 | 041 | 1 | 4 | 4 | 8 | 1 | 042 | 1 | 3 | 3 | 6 | 1 |
| 061 | 1 | 3 | 3 | 6 | 0 | 044 | 1 | 3 | 1 | 4 | 1 | 049 | 2 | 0 | 1 | 1 | 0 |
| 066 | 1 | 2 | 2 | 4 | 0 | 046 | 1 | 3 | 3 | 6 | 0 | 065 | 1 | 2 | 1 | 3 | 0 |
| 067 | 1 | 3 | 3 | 6 | 1 | 047 | 1 | 3 | 2 | 5 | 0 | | | | | | |
| | | | | | | 048 | 1 | 3 | 3 | 6 | 1 | | | | | | |

*2.2. Segmentation and Preparation (VascuCap vs. Slicer)*

We used an image processing software (vascuCAP Build A.3 25 January 2021 12:22:43; Elucid Bioimaging, Wenham, Mass) [22] to outline segments of the luminal and outer wall surfaces of the common and internal carotid arteries. After automatic segmentations, manual refinement was applied to correct the boundaries at the reconstructed surface in the lumen by two medical experts in consensus reading (ZM, ZC). The external carotid artery was excluded from the study after the bifurcation. The exclusion of the external carotid artery was realized by cutting it off, preferably with a surface perpendicular to the axis. After this semi-automatic segmentation and the vascular reconstruction had been performed, plaque geometry and tissue composition were automatically computed for the entire artery section. The segmented lumen data for each patient were extracted from the VascuCap software.

Manual segmentations were performed in an open-source software (3D Slicer, version 4.10.2. Kitware Inc., New York, NY, USA) by non-medical experts (BC, MBN) in consensus reading. Initial selection of voxels outlining the luminal segment were made by manually adjusting the lower and upper thresholds based on voxel density. Subsequently, each cross section was analyzed to include or exclude additional voxels by painting over or erasing, respectively. In order to check the viability of the manual segmentations, the three homogeneous groups were processed as follows.

The first patient group set was segmented individually by the medical (ZM, ZC) and non-medical experts (BC, MBN) without any corroboration from the other party (denoted as "Individual" in Figure 1). The second set was segmented collectively and at the same time with strong corroboration with the other party (denoted as "Collective" in Figure 1). After the corroboration the following conclusions were reached: 1. bifurcation segmentation has to be performed with more than two window settings; only using the automatic window setting is not satisfactory; 2. calcification should be manually reduced to compensate for the overestimation of the CTA; 3. circular calcification can conceal the contrast medium, and this artefact may give a false impression of occlusion or lead to under-segmentation of the lumen. The third set was segmented again individually without transferring any information on thresholding criteria or problematic vessel regions, but the conclusions of the collaboration were built into the methodology. Between the sessions, at least 2-week delay periods were held.

Initially, every image was segmented by both methods, then all the raw segmentations were loaded into 3DSlicer and the surface meshes were smoothed equally with a factor of 0.2. This step was needed for the later centerline calculation and cross-sectional analysis. Centerline calculations were performed by utilizing the Vascular Modelling Toolkits (VMTK) [23] framework. The initial centerlines were resampled and smoothed in VMTK to enable the proper calculation of curve-fitted perpendicular coordinate systems along the centerline. Based on these geometrical properties, the cross-sections were extracted along the centerline. Since the cross-sections are not circular, the equivalent diameters were calculated from the cross-sectional area. Although the cross-sectional area as an evaluation parameter can be sufficient, we chose the diameter because of a further evaluation step described later. The steps of the calculations can be seen in Figure 2.

Furthermore, the area between the curves is shaded according to whether the differences with respect to the segmentation performed in VascuCap are positive (dark red) or negative (navy blue). For the sake of convenience, shading is the same as in [24]. Thus, the phrases over-segmentation (light red shaded area) and under-segmentation (light blue shaded area) will be used if the diameter curve from Slicer has a positive or negative difference, respectively, compared to that from VascuCap.

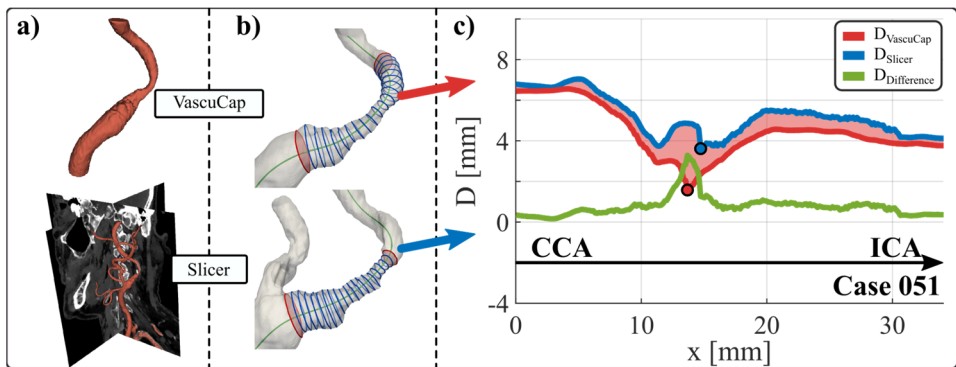

**Figure 2.** Steps of data preparation. (**a**) segmentation with both techniques (top VascuCap-semi-automatic segmentation by medical examiner, bottom Slicer—manual segmentation by non-medical examiner), (**b**) cross-section extraction and (**c**) diameter difference calculation along the centerline. The diameters are given in mm on the diagram. The dark blue line shows the lumen diameter segmented by Slicer, the dark red line shows the lumen diameter segmented by VascuCap and the green line shows the difference. ICA: internal carotid artery, CCA: common carotid artery.

### 2.3. CFD Analysis

The analysis was carried out with the packages of ANSYS 19.3. (ANSYS Inc). Since the semi-automatic segmentation did not include the external carotid artery section, the missing part was incorporated from the manual segmentation by merging the surface meshes with the CCA-ECA branch from the other geometry (See the geometries in Figures 5 and 6). Surface meshes were imported into SpaceClaim for geometry pre-processing before meshing. Unstructured numerical meshes of around 3.5 million cells were constructed including 8 prismatic inflation layers adjacent to the wall. The finite volume method was applied to solve the numerical problem in ANSYS CFX. The vessel wall was assumed to be rigid, and blood was modelled as a Newtonian fluid with a density and dynamic viscosity of 1055 kg/m$^3$ and 3.4 mPas, respectively [25]. A time-varying parabolic profile with velocity waveforms acquired from Doppler velocity measurements was prescribed as an inlet boundary condition on the CCA. Accordingly, the lengths of the heart rate-cycles became 0.783 and 0.929 s for the CS-010 and CS-042 cases, respectively, after averaging five periods from the measurements. Then, 5000 timesteps were defined for one heart rate-cycle as in [26] and 200 timesteps were exported from the solution of each cycle. Then, 0 Pa static pressure and no-slip boundary conditions were set for the outlets and vessel wall, respectively. The convergence criteria within the timesteps were set to be 10$^{-5}$. The simulations were carried out for three heart rate cycles but only the last period was used in the subsequent evaluation.

### 2.4. Evaluation

The reason behind using the calculated diameter instead of the area of the cross-sections for evaluation is that the diameter can be normalized by a unit of the voxel length. The first step was a qualitative analysis, which helps visualize the diameter difference curve normalized by the voxel difference (DDV curves) of all patients in each subgroup (in Table 2). The second step was the quantitative analysis of the geometrical parameters obtained by the semi-automatic and the manual methods in each patient group. Pearson correlation coefficients were calculated for the minimum diameter $D_{min}$ in the respective groups to compare the values of the semi-automatic and manual segmentations. Evaluation of $D_{min}$ was not applicable if the excluded external carotid artery was included in the most stenotic segment (CS-050, CS-004, CS-025, CS-029). In Table 3, the group-averaged $D_{min}$ values are presented. Correlations with the calcification metrics were also calculated. Here, patients with a calcification score of 0 (non-calcified soft plaque) were excluded from the calcification score correlation analysis as the calcification correlation evaluation was not applicable if the calcification score was 0 (CS-010, CS-031, CS-036). Here we want to remark

that in the calculation of the average voxel differences, small segments, where the external carotid artery cut-out (at the bifurcation point) in VascuCap introduced a non-applicable difference were not taken into account. In the third step, the effect of the calcification severity on the difference in each segmentation group with the two methods was correlated. Table 4 summarizes the effect of calcification on the variance of segmentation between examiners, methods and homogeneous patient groups. Significance was calculated with paired Student t-test and the significance level was set at 5%.

CFD simulations were performed on two patients to demonstrate how subtle differences in geometry can affect the hemodynamic flow field. Each patient was simulated with both geometries generated by the semi-automatic and manual segmentation. In Figure 5 the locations of the data extraction are visualized in the middle, while on the right, diagrams of velocity profiles along a normalised diameter length are gathered at three locations along the centerline (P1, P2, P3). Systolic and diastolic time instants were defined as the highest and lowest velocities, respectively. Care has to be taken here to emphasize the non-dimensionalization of the diameter. We wanted to project these velocity profiles onto the aforementioned reference to capture whether the velocity profiles are scale-dependent, or not.

### 3. Results

#### 3.1. Segmentation Comparisons

The equivalent diameter curves of both segmentations are plotted along the centerline and their differences can be visualized between the semi-automatic and manual segmentations. These difference curves are depicted for each case in Figure 3. An average diameter difference normalized by the voxel length (ADDV) from the aforementioned diameter difference curve is computed for all patients. Group averages and standard deviations were also computed to compare the outcome between the groups. Furthermore, the value of the minimum diameter ($D_{min}$) was extracted for all the patients and averaged for each group.

The summary of the evaluation can be seen in Figure 3 for all the cases. In the left panel of Figure 4 only the diameter difference curves are shown, normalized by the voxel dimension (DDV curves). These diagrams collectively show the cases in each group for visual interpretation. By visual inspection an excessive over-segmentation can be seen in the first group by the manual segmentation. In Group II difference curves tend to move closer to the 0 value with some cases of under-segmentation. Visually, the cases of Group III. are similar to those in Group II. with less under-segmentation. In the right panel of Figure 4 a Bland–Altman diagram can be seen to analyze the average diameters obtained from both methods that comprises all three groups. The quantitative results confirm the visual interpretations closely. Furthermore, it shows that the cases in Group II. are less scattered around the overall average than in Group III.

Group averages were computed with weighted averaging by the centerline length. As it can be seen from the quantitative results of the ADDV measurements in Table 2, the largest mean value of ADDV and corresponding standard deviation (std) was in Group I. with 1.16 and 1.33 respectively. The mean values in Group II. and III. are 0.65 and 0.75, while the deviations are 1.09 and 0.91, respectively. The average for the present whole study population was 1.17 with a standard deviation of 1.12.

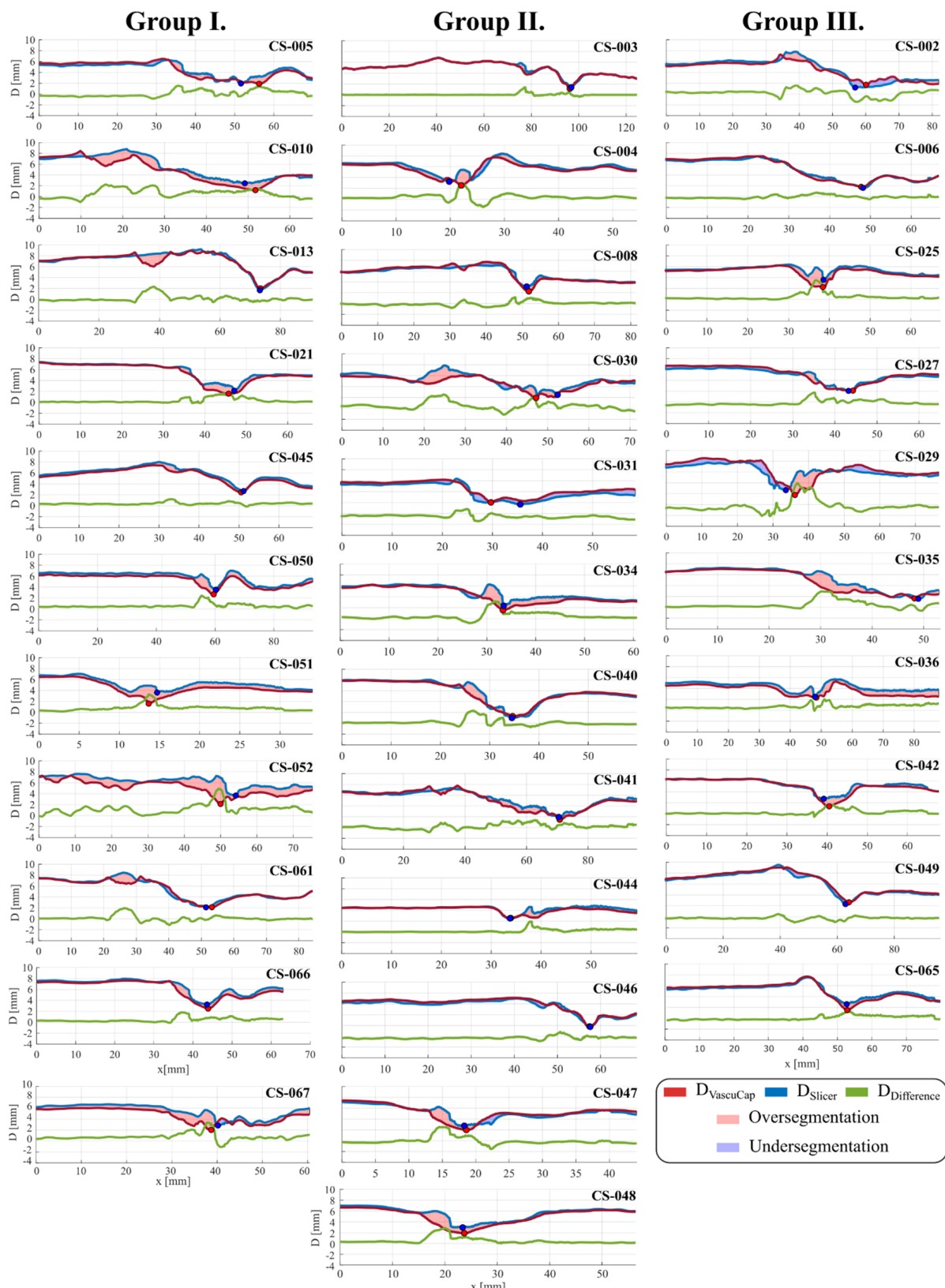

**Figure 3.** All evaluated cases as described in Figure 2.

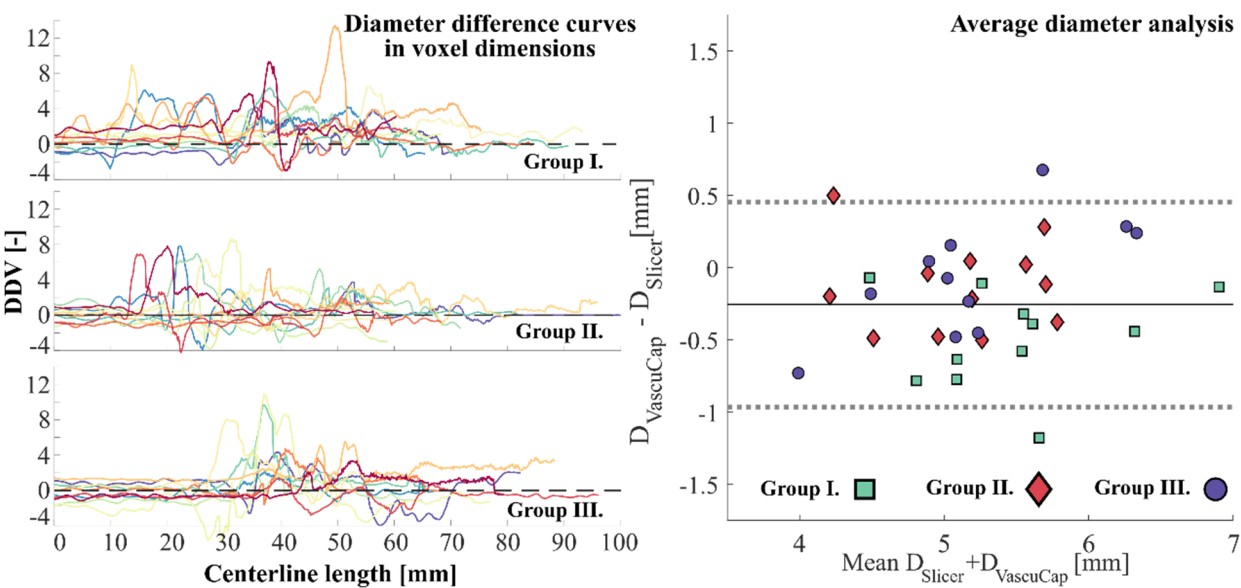

**Figure 4.** Left: Diameter difference curves normalized by the voxel dimension (DDV curves). Each subplot represents the collection of all cases within the respective group. Different colors were used only to distinguish between the cases. The left and right parts correspond to the beginning of the reconstructed CCA and ICA sections, respectively. Right: Bland–Altman plot on the analysis of the average diameters. ICA: internal carotid artery, CCA: common carotid artery.

**Table 2.** Top: Average diameter difference in voxel dimensions (ADDV) for each case. Bottom: Mean and Std. values for each group. The cases highlighted with orange shading were excluded from the calcification analysis due to lack of any calcification.

| Group I.—Individual | | Group II.—Collective | | Group III.—Separated | |
|---|---|---|---|---|---|
| **ID** | **ADDV [–]** | **ID** | **ADDV [–]** | **ID** | **ADDV [–]** |
| 005 | 0.19 | 003 | 0.11 | 002 | 0.26 |
| 010 | 1.73 | 004 | 0.79 | 006 | 0.12 |
| 013 | 0.37 | 008 | 0.06 | 025 | 0.82 |
| 021 | 0.88 | 030 | 1.31 | 027 | 0.42 |
| 045 | 0.95 | 031 | 1.36 | 029 | 1.77 |
| 050 | 1.35 | 034 | 1.33 | 035 | 0.29 |
| 051 | 2.73 | 040 | 0.32 | 036 | 1.99 |
| 052 | 2.14 | 041 | 0.59 | 042 | 0.64 |
| 061 | 0.30 | 044 | 0.54 | 049 | 0.65 |
| 066 | 1.01 | 046 | 0.77 | 065 | 0.20 |
| 067 | 2.11 | 047 | 0.35 | | |
| | | 048 | 0.92 | | |
| **Mean** | **Std.** | **Mean** | **Std.** | **Mean** | **Std.** |
| 1.16 | 1.33 | 0.65 | 1.09 | 0.75 | 0.91 |

Results obtained for $D_{min}$ are collected in Table A1 of the appendix (Appendix A). In Table 3, the summary of the data set is given for the analysis on the minimum diameters. The results show that the group average $D_{min}$ value in each segmentation set is larger for the manual segmentation. The relative difference and deviation are the highest in the first, entirely individual segmentation group. The smallest relative difference was measured in the last group with 9.8% but with almost the same deviation (36.47%) as in the first group. The relative difference was 19.19% in the collective segmentation group with the lowest deviation 27.92%. The Pearson correlation analysis revealed that moderately strong (r = 0.68) and significant ($p < 0.05$) correlation was found between the manual and semi-

automatic segmentation in the collective group. Additionally, significant ($p < 0.01$) although weak correlation (r = 0.138) was found in the first individual group. Non-significant and almost zero correlation was obtained in the separated segmentation group.

**Table 3.** Results of the comparison of group averaged $D_{min}$ values between the two segmentation methods.

|  | $D_{min\ AVG}$ VascuCap | $D_{min\ AVG}$ Slicer | Relative Difference | Relative Std | Correlation | $p$ |
|---|---|---|---|---|---|---|
|  | [mm] | [mm] | [%] | [%] | [–] | [–] |
| Group I. | 1.950 | 2.636 | 35.16 | ∓38.75 | 0.138 | <0.01 |
| Group II. | 1.815 | 2.178 | 19.99 | ∓27.92 | 0.680 | <0.05 |
| Group III. | 1.883 | 2.067 | 9.80 | ∓36.47 | 0.084 | <0.25 |

### 3.2. Effect of Calcification on Correlation of Segmentation

ADDV was correlated to three calcification metrics within each segmentation group, independently. The associated values are listed in Table 1 (calcification metric data) and in Table 2 (ADDV data). Cases listed in Table 2 by an orange background color were not taken into account for the calculations, as the calcification score was zero for these cases and therefore, we considered them inapplicable in this analysis. The calculated data of the correlation analysis are collected in Table 4. Significant results were obtained for all the calculations. In the individually segmented group, moderate correlations (r = 0.59; 0.44; 0.61) were found between ADDV and the calcification metrics. Moderate and strong correlations (r = 0.71; 0.51; 0.82) were revealed for Group III. Lastly and most interestingly, a mostly weak (0.19; −0.20; 0.46) correlation was found for the collective Group II.

**Table 4.** Pearson correlation coefficients and $p$ values calculated between ADDV (average diameter difference normalized by the voxel length) and calcification metrics in each group.

| ADDV vs. |  | Group I. Individual | Group II. Collective | Group III. Separate |
|---|---|---|---|---|
| **Calcification Score** | Corr. Coefficient | 0.59 | 0.19 | 0.71 |
|  | $p$ | <0.001 | <0.0001 | <0.001 |
| **Calcification Extent** | Corr. Coefficient | 0.44 | −0.20 | 0.57 |
|  | $p$ | <0.05 | <0.001 | <0.01 |
| **Calcification Thickness** | Corr. Coefficient | 0.61 | 0.46 | 0.82 |
|  | $p$ | <0.01 | <0.001 | <0.001 |

The collective segmentation results in an increased similarity for the segmented geometry, independently of the calcification severity. On the other hand, separated segmentation, even if there was consensus about the segmentation rules, was affected strongly by the calcification severity metrics.

### 3.3. Effect of Differences of Segmentation on Flow Analysis

The overview of the results obtained from the CFD simulations can be seen in Figures 5 and 6. Velocity profiles were extracted at three locations and time-averaged WSS (TaWSS) was calculated by cycle-averaging the last cycle of the simulation.

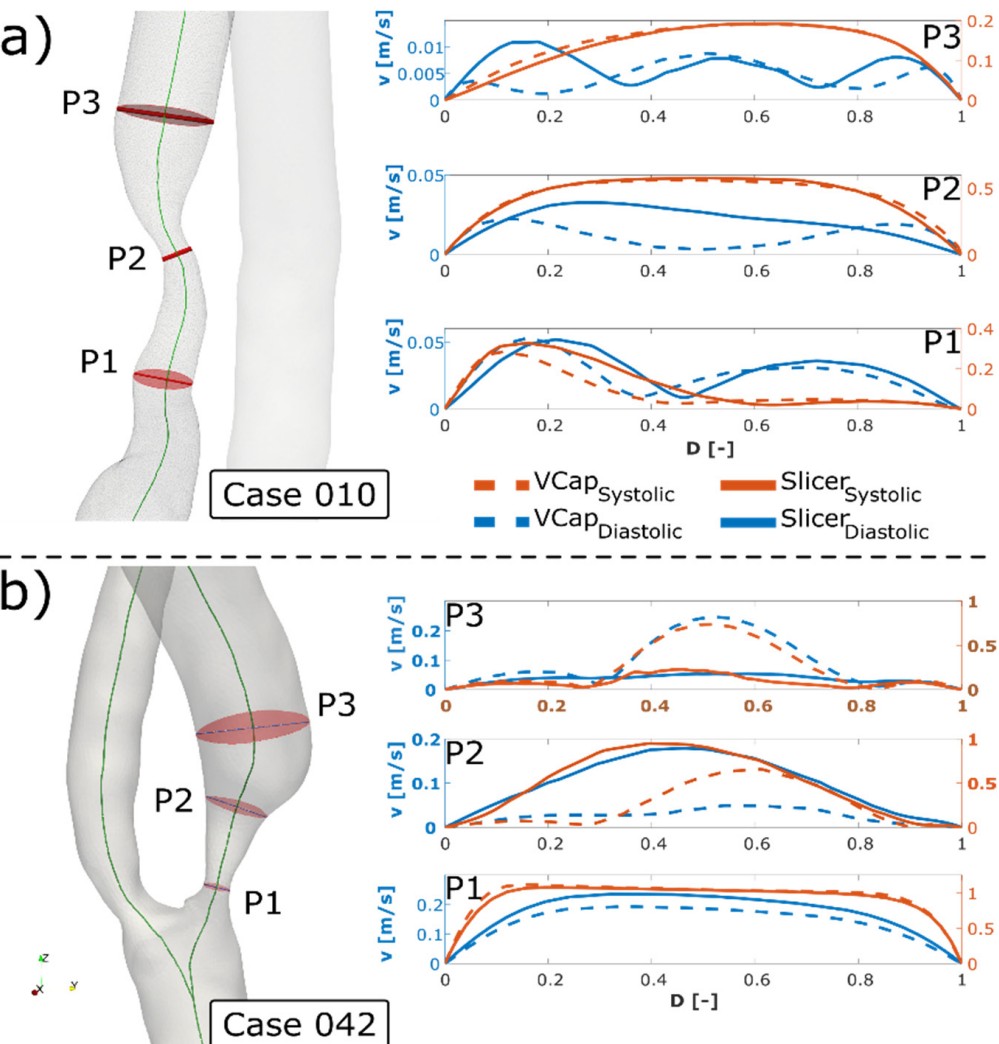

**Figure 5.** Results of the CFD simulations for Case 010 (**a**) and Case 042 (**b**). Velocity profiles extracted in the systolic and diastolic time instants are colored orange and blue, respectively. Dashed and continuous lines correspond to the VascuCap (semi-automatic) and Slicer (manual) geometries. The profiles are plotted along the non-dimensional diameter length. CFD: Computational fluid dynamics.

The vicinity of the stenosis and the three locations can be seen on the left of Figure 6 for cases 010 and 042. At the systolic time instant, the velocity profile at the most stenotic location looks similar for both cases with small to negligible absolute differences. On the other hand, in the diastolic phase, qualitative differences are present as the shape of the profile varies. Other locations for both cases show similarities in the systolic phase that the velocity profiles look similar. However, major differences can occur in the diastolic phase, such as at the location P3 in Case 010 and locations P2 and P3 in Case 042.

Figure 6 demonstrates that the TaWSS field near the stenosis globally looks similar, yet clear qualitative differences can be visualized between the semi-automatic and manual segmentation results, even in this cycle-averaged representation. Quantitatively, for the maximum TAWSS, 30% difference was measured for Case 010 (ADDV = 1.73), while for Case 042 (ADDV = 0.64) this difference can drop to 4%. Further evaluation was performed by mapping of the TaWSS distribution to a parametrized rectangular space [27] (illustrated as flattening in the middle panel of Figure 6). This space is suitable for comparisons, thus values on two lines were extracted which is essentially the comparison of two circumferences at different altitudes from the bifurcation. Both of them show that at the stenosis (Line 2), the values—obtained from the geometry created by

manual segmentation—considerably underestimate those obtained by the semi-automatic segmentation.

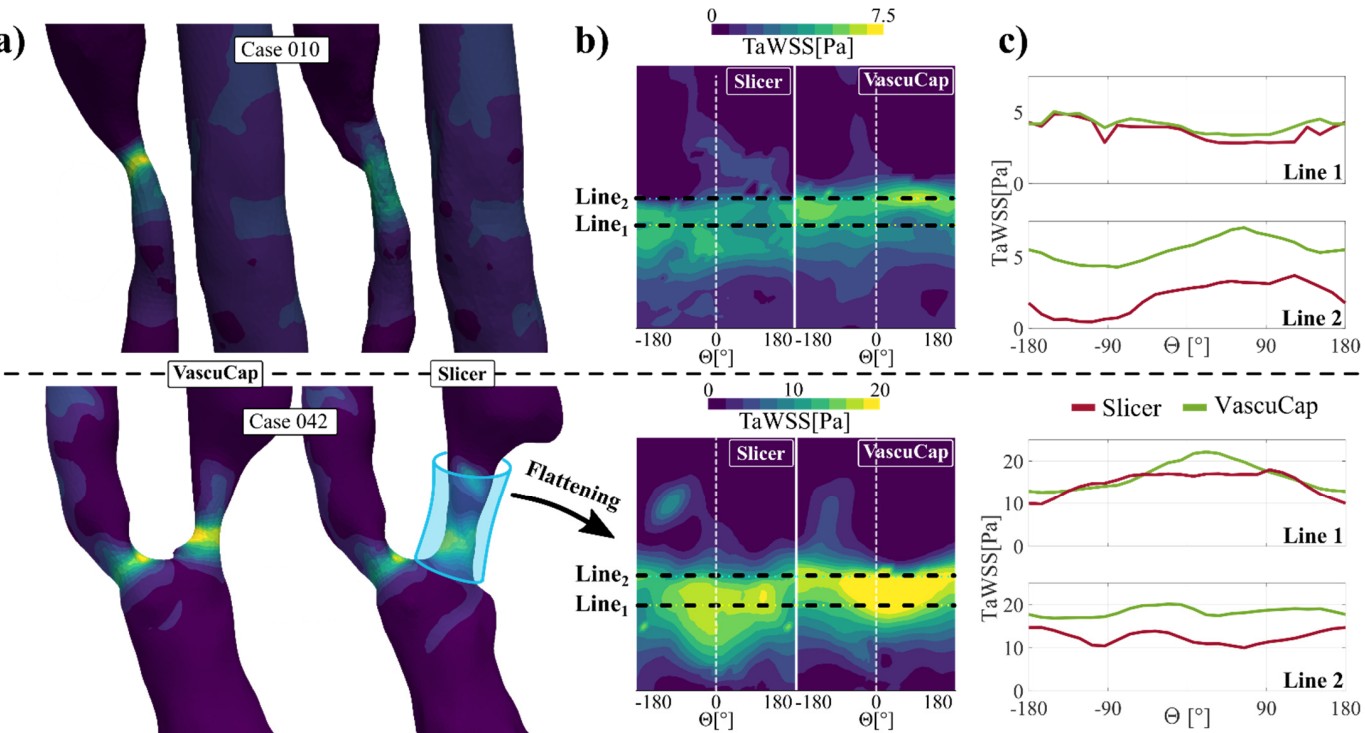

**Figure 6.** Results of the CFD simulations for Case 010 and Case 042. (**a**) Time-averaged WSS field near the stenosed vessel segment visualized on the vessel wall. (**b**) Flattened TaWSS maps near the stenosis for easier comparison. (**c**) circumferential TaWSS values at different altitudes (Line$_{1,2}$) from the bifurcation point; WSS: wall shear stress TaWSS: Time-averaged WSS. CFD: Computational fluid dynamics.

## 4. Discussion

### 4.1. General Discussion

One of the main objectives was to investigate the respective segmentation capabilities of an experienced medical and a non-medical investigator. There are no previous data about the difference between medical versus non-medical investigators in the case of carotid artery lumen segmentation. In our analysis, the medical examiners performed the segmentation in consensus reading, which could be a limitation of our study. ACRIN (American College of Radiology Imaging Network) indeed emphasizes that potential variability among observers requires the same level of attention as potential variability among study subjects [28]. The consensus interpretation in imaging research was classified as a limitation for study by [29], however, in our study design, not the interobserver variability, but rather the variability due to the different segmentation methods was the aim of the investigation. The intraobserver variability by remodeling evaluation was published in the original articles [21].

The two software tools, which were used in this study, were developed for different purposes, but both are unique in their special fields. Currently, VascuCap has the FDA approval for a clinical research setting. However, there is no carotid plaque analysis software for clinical use with FDA approval, although high correlation and low bias have already been published between the in vivo software image analysis and ex vivo histopathologic quantitative measures of atherosclerotic plaque tissue characteristics, as well as low reader variability and high sensitivity for changes in plaque characteristics [22,29]. Carotid plaque geometry and tissue composition have been proven to be measured reliably from clinical CTA images using the VascuCap software. A minimum change, 4% was detectable in

plaque volume when the same observer performed both measurements, but it increased to 10% for different observers [30]. There is no published data about lumen segmentation alone, only about the plaque volume (lumen + plaque burden) with VascuCap. Similar observations were made as in the MATCH study, in the case of manual segmentation mostly over-segmentation was found [24].

### 4.2. Effect of Differences of Segmentation

The corroboration leads to some key conclusions. There were several habitual differences between the medical and non-medical examiners in segmentation processing and the choice of setting options. Some cornerstones, listed in the methods section, were established by the study working group to reduce the differences. Our aim was to reduce the differences attributed to the different experiences in segmentation and to measure the difference in lumen geometry between the two segmentation methods in the third (separated) group. The results show that over- and under-segmentation can be significantly reduced after the definition of cornerstones, but the differences caused by severe calcification can be reduced only with corroboration during the segmentation. The quantitative analysis revealed that the ADDV average and the standard deviation of the studied patient population were lower than what we measured in the first individual segmentation group and considerably larger than in the second and third groups. This result can indicate the usefulness of a collaboration between the medical and non-medical examiner, during the geometry preparation for a computational study.

Manual thresholding might bias the risk of over/under-segmentation but elaborating on some cornerstones can decrease the associated errors. A larger risk of over/under-segmentation can occur locally if a high amount of calcification is present, for two reasons. Calcification blooming can easily distort the image locally, segmentation can be labor-intensive and mostly subjective, and manual oversight might cause/result differences at these locations. Another source of bias can originate from the shadowing effect of complex-shaped calcification. Sometimes the contrast medium may not be present at some almost enclosed lumen pockets.

The external carotid artery exclusion could be a source of potential bias if several observers perform the lumen segmentation instead of consensus reading. Advanced semi-automatic imaging analysis protocols are on the horizon of daily clinical practice. However, the main advantage of semi-manual segmentation is the automatic centerline setting and the automatic modification of the centerline after manual correction, which can spare time for the examiner, but the external carotid artery exclusion plane is also automatically modified by the software. The manual segmentation allows the external carotid artery exclusion perpendicularly to the axis of the artery, and the centerline and minimum lumen diameter settings can be defined more precisely if the standardized window setting is applied.

Complex stenotic lesions with severe calcification are challenging even for experienced investigators. A previous study that compared automatic versus manual segmentation in stenotic carotid artery lesions, showed some bias between the the observers andthe automated method. The differences between the automated and the manual method and also between the observers were significant for calcium volume [31]. The previously published articles differ in the HU threshold setting of calcifications (over 96—over 250 HU) for carotid artery plaque analysis studies based on CT angiography images [32–35]. These notable discrepancies in the literature show that a wide and dynamic window setting must be used to precisely evaluate the lumen segmentation at the calcified part of the target vessel. Different plaque component analysis studies use different CT angiography settings and require different image quality, so calcification definitions based on only HU units can cause severe bias among studies. Furthermore, circular calcification can cause an artificial loss of automatically detectable contrast agent in pre-occlusive severe stenotic lumens. Due to frequent deletions of the residual lumen in calcified plaques, high-grade stenosis might be overestimated as occlusions, according to a previous study about automatic

segmentation [19]. In the case of pre-occlusive, severely calcified lesions, the automatic centerline identification is impossible or underestimates the lumen diameter due to the artificial shadowing of the contrast agent caused by circular calcification.

The minimum diameter is a local parameter and with increasing severity of calcification it is progressively harder to get it properly by manual segmentation. This is shown by the summarizing Figure 3 and the left panel in Figure 4. (DDV curves). In Group II, when we worked together, care was taken for the most problematic sections and this in turn improved the estimation of minimum diameter. Although the correlation in Group III is not significant because of the low number of available patients, our rationale is the following. Due to the collaboration in Group II, we have a strong correlation for the minimum diameter as we could verify each other's work in person. This idea complements the results of the correlation analysis for the calcification metrics since when we worked together there was a significantly poor correlation between the metrics and ADDV values. This is one of the most important results of the study. The lack of correlation between calcification and average difference (ADDV) is due to the improved quality of the collaborative segmentation: none of the participants displays a bias in any direction because of the calcification. In contrast, the results for the other two groups showed a good correlation with the calcification metrics and a poor correlation for the minimum diameter, because of the inversion of the above argumentation. The main difference between Group I and Group III is that the overall segmentation improved because of what had been learned during the collaboration phase.

### 4.3. Effect of Differences on the Flow Field

The exploratory CFD simulations showed that the WSS field can be significantly affected by differences in the segmentation. The velocity field is much less prone to the presence of small differences (at the smallest diameter), but moderate differences that weakly change the morphology of a certain vessel section can change the velocity field considerably as was similarly demonstrated by others [36,37]. Our results for the TaWSS field show the effect of how a more localized difference (Case 042) could alter the results even qualitatively and how a broadband difference could blur the WSS field. Automatic segmentation can spare time and effort for the examiner, but the clinically relevant stenotic and post-stenotic part of the target vessel should be prepared manually, even if it is time-consuming. Our results highlight the importance of the experienced medical investigators and the co-working of experts from different fields. Thus, knowledge transfer with a medical practitioner is essential for the non-medical examiners to construct lumen segmentations that are applicable for CFD analysis without major inconsistency in the critical stenotic segments. The CFD analysis evaluation needs special skills to substantiate appropriate results, proper settings of the analysis are essential [18,38] beside the knowledge of the physiologic and pathologic processes.

### 4.4. Limitations

A possible limitation of our study is that the manual vs. semi-automatic and the medical vs. non-medical user's segmentations were not clearly separated, rather manual segmentation was associated with the non-medical and the semi-automatic segmentation with the medical examiner.

The reason for that was that the cooperation seen between medical practitioners and research engineers has its own laws. The software used by medical personnel are tailored for diagnostic purposes (such as VascuCap in our study) and their detailed working mechanisms are not known—they are rather black boxes. On the other hand, they are easy to use. Engineers use different software (3D slicer, MeshLab, etc.) whose usage is more labor-intensive and needs some form of special training, but they give full control for the user over the geometry manipulation.

In everyday life it a physician never uses engineering segmentation software, or vice versa. Although from a strict methodological point of view it may have been necessary to

try cross usage, it would have lacked any relationship to reality. Our study emphasizes that even though today it is not a challenge to create a geometry out of medical images for simulations, caution has to be exercised on how to construct those geometries.

A further limitation is the low number of cases in the three segmented homogenous patient groups. CFD simulation may have been performed in more cases, but its purpose was to demonstrate the possible consequence of "even a single voxel" difference, not to calculate the its exact effect.

## 5. Conclusions

Our exploratory study aimed at investigating how to construct adequate computational geometries for blood flow simulations from CTA images that are appropriate for comparison with a plaque image analysis using commercially available software. The study shed some light on important elements on how patient-specific computational studies should be initiated.

Our study entails a recommendation for a tighter collaboration in the phase of geometry creation, as our results revealed a better outcome in several aspects. In the case of severely calcified lesions it is particularly important that they are segmented under medical supervision or in consensus reading with a medical practitioner similarly to the method of Group II. A collective in-depth review of the medical images should precede a set of manual or automatic segmentations before using them in Computational Fluid Dynamics simulations to ensure a solid basis for proper comparison with plaque imaging analysis. Thus, one of the closing arguments of our study—if ever simulation would replace some of the clinical studies—was to show that coworking is much more reliable compared to individual parallel working groups in the case of simulation-based findings in clinical settings. Non-significant differences in the segmentation can lead to significant differences in the computed flow field, thus we suggest that lumen segmentation for CFD analysis should be performed after a special training with manual or semi-automatic software.

**Author Contributions:** Conceptualization, Z.M., B.C. and P.S.J.; methodology, Z.M. and B.C.; software, Z.M., B.C., Z.C. and M.B.N.; validation, Z.C., M.B.N. and G.P.; formal analysis, B.C. and G.P. investigation, B.C. and Z.M.; resources, Z.M., Z.C., B.C. and M.B.N.; data curation, B.C., M.B.N., G.P. and Z.M.; writing—original draft preparation, Z.M., B.C., Z.C., M.B.N., G.P. and P.S.J.; writing—review and editing, P.S.J., G.P. and G.H.; visualization, B.C. and M.B.N.; supervision, P.S.J., G.P. and G.H.; project administration, Z.C.; funding acquisition, P.S.J. All authors have read and agreed to the published version of the manuscript.

**Funding:** Project no. NKFI-K129277 ("Evaluation of cerebrovascular events in patients with occlusive carotid artery disorders based on morphological and hemodynamic features") has been implemented with the support provided by the Ministry of Innovation and Technology of Hungary from the National Research, Development and Innovation Fund.

**Institutional Review Board Statement:** The study was conducted according to the guidelines of the Declaration of Helsinki and approved by the Institutional Review Board (or Ethics Committee) of Semmelweis University (84/2019 10 May 2019).

**Informed Consent Statement:** Informed consent was obtained from all subjects involved in the study.

**Conflicts of Interest:** The authors declare no conflict of interest.

## Appendix A

**Table A1.** Minimum diameter $D_{min}$ values (in mm dimension) obtained from both segmentation methods for all cases.

| Group I.—Individual | | | Group II.—Collective | | | Group II.—Separated | | |
|---|---|---|---|---|---|---|---|---|
| $D_{min}$ | VascuCap | Slicer | $D_{min}$ | VascuCap | Slicer | $D_{min}$ | VascuCap | Slicer |
| *005* | 1.893 | 1.946 | *003* | 1.146 | 1.384 | *002* | 1.709 | 1.199 |
| *010* | 1.189 | 2.457 | *008* | 2.268 | 3.133 | *006* | 1.814 | 1.614 |
| *013* | 2.029 | 1.660 | *030* | 1.939 | 2.516 | *027* | 2.092 | 2.040 |
| *021* | 1.592 | 2.151 | *031* | 2.045 | 1.665 | *035* | 1.628 | 1.538 |
| *045* | 2.438 | 2.670 | *034* | 1.538 | 2.363 | *036* | 2.308 | 2.477 |
| *051* | 1.577 | 3.614 | *040* | 1.289 | 0.961 | *042* | 1.336 | 2.722 |
| *052* | 2.130 | 3.668 | *041* | 1.488 | 1.943 | *049* | 2.676 | 2.349 |
| *061* | 2.137 | 2.119 | *044* | 2.559 | 2.526 | *065* | 1.502 | 2.600 |
| *066* | 2.484 | 3.229 | *046* | 1.824 | 1.755 | | | |
| *067* | 2.036 | 2.846 | *047* | 1.973 | 2.763 | | | |
| | | | *048* | 1.898 | 2.952 | | | |

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
