# Peer review of "Comparison of Manual versus Semi-Automatic Segmentations of the Stenotic Carotid Artery Bifurcation"

_applsci, doi:10.3390/app11178192_

Round 1
Reviewer 1 Report
First of all congratulation on your effort. I wonder however why is this study so "anatomical" - which means to me it completely is not. I would rather change the key words to make the manuscript more specific, otherwise it is qualified to typical anatomical manuscripts.
Secondly if you wish to make it more anatomical use some references that reflect the problme from anatomical stanpoint. I have selected four different articles dealing with anatomy of the carotid bifurcation. Maybe you can use any?
Folia Morphologica 2021
Reference luminal diameters of the carotid arteries among healthy Nigerian adults
- P. K. Kpuduwei, E. K. Kiridi, H. B. Fawehinmi, G. S. Oladipo
Ahead of Print
Internal carotid and vertebral arteries diameters and their interrelationships to sex and left/right side
- Spasojević, S. Vujmilović, Z. Vujković, R. Gajanin, S. Malobabić, N. Ponorac, L. Preradović
Folia Morphologica Vol 79, No 2 (2020)
Abstract · PDF
Variations in carotid sinus anatomy and their relevance to carotid interventions
- T. West, C. Brassett, M. E. Gaunt
Folia Moprhologica Vol 77, No 4 (2018)
Morphological variation of carotid artery bifurcation level in digital angiography
- Kurkcuoglu, C. Aytekin, H. Oktem, C. Pelin
Folia Morphologica Vol 74, No 2 (2015)
Author Response
Dear Reviewer,
Thank you for your review and useful comments. We have corrected some tiny misspellings in the text. The word “anatomical” was deleted from the keywords. We have added an explanation to the anatomy of the carotid bifurcation using reference from your suggested list and we have done a minor change in the text. the manuscript with the changes has been uploaded, please, see the attachment. We hope with these minor changes the manuscript can be published. We are looking forward to hearing your comments.

Reviewer 2 Report
The authors demonstrate qualitative and quantitative differences in manual and semi-automatic segmentation of stenotic carotid artery bifurcations imaged by computed tomography angiography (CTA). These differences depend upon the presence or absence of collaboration between the non-medical and medical investigators performing these tasks, experience working together (i.e. previous collaboration), and degree of lesion calcification. The differences in geometry resulting from these segmentations can result in meaningful differences in computational fluid dynamic (CFD) simulation results.
While the manuscript does offer interesting insights to the field to guide future research relying upon image reconstruction and image-based modeling, there are some fundamental concerns regarding the aims, methods, results, and conclusions of the paper. While not necessarily insurmountable, these issues need to be addressed through further clarification, additional analysis, and/or more extensive discussion of study limitations. I provide my comments according to section below (indicating lines in the manuscript), though there is some repetition and common themes throughout. I mark two comments with asterisks (*) which I find to reflect my most serious concerns with this work.
ABSTRACT
You indicate that “the outcome of the blood flow simulation may vary because of a single voxel difference in the stenotic lesions” (13-14). While this is, strictly, true – simulation results will vary to some extent – changes by a single voxel are unlikely to yield meaningful differences in all but the most extreme cases, making this claim overstated.
You seem to suggest (13-14) that changes in blood flow simulation can lead to conflicting clinical findings, yet I’m not aware of CFD results of single bifurcations being used clinically – only in research contexts. (If you are referring to FFRCT, which models the entire coronary tree, then the claim that a single-voxel difference will change clinical findings seems even more far-fetched.)
MATERIALS AND METHODS
While “CFD” is defined in the text, the definition is missing from the Figure 1 caption (108).
The authors report that “the patients lacking the highest quality images…were excluded, since a high arterial contrast enhancement is advantageous for accuracy” (116-118). While this was a reasonable action, it would also be interesting to explore the impact of image quality on segmentation variability between methods. (While I believe this could contribute to the findings of the paper, the authors should not feel compelled to conduct this study if the work is prohibitive – this is strictly a suggestion.)
In reporting the characteristics of the homogeneous groups, consider whether a plot showing distribution of each variable in each group would be more effective than presenting the information as a table (Table 1; 130-132).
In describing the CFD analysis and the incorporation of the external carotid artery, the authors reference Figure 4 (191), which does not seem relevant in this context.
In describing the evaluation of results, the authors indicate that “the second step was the qualitative analysis of the geometrical parameters” (212). Did the authors instead intend to indicate that they carried out quantitative analysis, as suggested by the calculation of Pearson correlation coefficients?
The order of the Tables is confusing. In particular, why are Tables 3 and 4 introduced before Table 2 (first introduced on lines 217, 226, 263, respectively)?
*“In this paper, a comparison is performed between the manual and semi-automatic segmentations performed by non-medical and medical investigators, respectively” (16-18). Through this approach, you introduce two confounding factors: (1) method of segmentation (manual vs. semi-automatic); (2) training of personnel (non-medical vs. medical). As such, your methodological approach – without further work – does not readily allow for distinction between the influence of each of these influential factors. How do you decouple these factors? The authors emphasize in the discussion that “one of the main objectives was to investigate the respective segmentation capabilities of an experienced medical and a non-medical investigator” (330-331), yet they haven’t precisely done so – this would have required the two groups of investigators to use the same method of segmentation. Elsewhere in the discussion, the authors note that “the variability due to the different segmentation methods was the aim of the investigation” (339-340), yet they haven’t precisely shown this either – this would have required the same group of investigators to use the different methods of segmentation. The method is inconsistent with the stated aims – which are themselves inconsistent – as well as the claims and conclusions.
RESULTS
The interchangeable use of “semi-automatic”/“manual” and “medical expert”/“non-medical expert” (e.g., 240), is somewhat confusing. See also comments above regarding confounding factors.
Figure 4 (257) is not effective. The authors are strongly encouraged to find a different way to convey whatever information they wish for the reader to glean from these results. This could, perhaps, take the form of a scatter plot and/or Bland-Altman plot comparing the diameters (though other effective forms are certainly possible).
A minor semantic issue – the authors direct the readers to “the supplementary materials” (272), though Table A1 is instead found in the appendix (Appendix A). This led to some confusion as I searched unsuccessfully for a supplementary materials section until I stumbled across the appendix at the end of my review.
When discussing the effects of differences of segmentation on flow analysis, the authors refer to Figure 4 (307-308), though I am quite confident that they instead meant to reference Figure 5.
*In reporting the effect of differences on segmentation on flow analysis, the authors indicate percent differences in maximum time-averaged wall shear stress (TAWSS), then insist that “further quantitative analysis is inapplicable here, as qualitative analysis can describe the differences well” (314-315). I strongly disagree. You earlier state that “the aim of our project is computational geometry reconstruction for blood flow simulations to make it suitable for comparison with plaque image analysis…” (14-16). If your aim is to correlate CFD results with plaque distribution, then the spatial distribution of TAWSS – not simply maximum TAWSS – is critically important, making further quantitative analysis highly applicable here. The two surfaces should be mapped to each other in order to compare values throughout the vessel, rather than just comparing the maximum of each (which may not even occur at the same location).
Results are reported for velocity profiles “at the systolic time instant” (316). Systole is typically defined as the period over which the heart is ejecting blood – how did you define or select a single “systolic time instant”? (The same applies for how the timing for the diastolic velocity profile was selected.)
The velocity values reported in Figure 5 (322) seem unrealistically low. Typical internal carotid artery velocities are 0.4 m/s (diastole) to 1.25 m/s (systole) in normal areas and 1-2.3 m/s (diastolic-systolic) in highly stenosed areas, yet you report values all less than ~1 m/s (and, most concerning, less than 0.5 m/s during systole in the stenosed region, P2, of Case 010). How do you explain this discrepancy?
DISCUSSION & CONCLUSION
You seem to suggest that Group III yielded preferable outcomes to Group I. However, Group III had “non-significant and almost zero correlation” in minimum diameter (282; Table 3), and “moderate and strong correlations” between average diameter difference and calcification metrics were found for this group (295-296; Table 4). First, further explanation of how correlation decreased alongside relative minimum diameter difference is warranted, as this is an unexpected and peculiar observation. Second, how do you justify recommending the course of action resulting in Group III’s results when outcomes seem so poor?
Author Response
Answers for the Reviewer 2 comments
Dear Reviewer,
Thank you for the detailed review, it helps us a lot in revising the text and properly explain our findings. The comments to the abstract are considered in the main text. The Methods, Results and Discussion sections were corrected to clearly interpret the questioned parts. Additional paragraphs, figures and results were added to the manuscript to enhance the understanding of the concerned parts. The detailed answers are below in Italic style:
- ABSTRACT
You indicate that “the outcome of the blood flow simulation may vary because of a single voxel difference in the stenotic lesions” (13-14), thus the mean single voxel difference, which is along the lumen, can cause serious misinterpretation in case of 3D structures containing the critical slices (which represent the stenotic area in 3D) . While this is, strictly, true – simulation results will vary to some extent – changes by a single voxel are unlikely to yield meaningful differences in all but the most extreme cases, making this claim overstated.
Thank you for pointing out this inaccuracy in our abstract. This part of the abstract was not precisely formulated in view of our results. The abstract is changed accordingly. A more elaborate explanation is given here and in the Discussion section.
In the abstract, the meaning of the voxel differences was misleadingly formulated. Here we meant not only one voxel difference, but a mean voxel difference along the whole examined vessel section. Even a single mean voxel difference can alter the flow field noticeably since it effectively alters the diameter and thus the cross-sectional area of the vessel. If a larger portion of the source of the differences is located at and around the stenotic section the (still with a mean difference of one voxel) alteration in the flow field can be substantial.
You seem to suggest (13-14) that changes in blood flow simulation can lead to conflicting clinical findings, yet I’m not aware of CFD results of single bifurcations being used clinically – only in research contexts. (If you are referring to FFRCT, which models the entire coronary tree, then the claim that a single-voxel difference will change clinical findings seems even more far-fetched.)
Indeed, we used the word clinical findings. We replaced it with any other research findings.
- MATERIALS AND METHODS
While “CFD” is defined in the text, the definition is missing from the Figure 1 caption (108).
The missing definition has been inserted.
The authors report that “the patients lacking the highest quality images…were excluded, since a high arterial contrast enhancement is advantageous for accuracy” (116-118). While this was a reasonable action, it would also be interesting to explore the impact of image quality on segmentation variability between methods. (While I believe this could contribute to the findings of the paper, the authors should not feel compelled to conduct this study if the work is prohibitive – this is strictly a suggestion.)
Exploring the impact of image quality on segmentation variability between methods is a very interesting new possible research direction. It would, however, be beyond the scope of this paper
In reporting the characteristics of the homogeneous groups, consider whether a plot showing distribution of each variable in each group would be more effective than presenting the information as a table (Table 1; 130-132).
Thank you for the suggestion, we have generated a figure with a histogram plots for the calcification score. We think that even the calcification score (comprising the calcification extent and thickness) could not show the distribution well. The other variables by definition are not suitable to be presented in a histogram form. Thus, the variables remained in a table format as we think that it is more suitable to present these data.
In describing the CFD analysis and the incorporation of the external carotid artery, the authors reference Figure 4 (191), which does not seem relevant in this context.
Unfortunately, the figure numbering was wrong, we have renumbered the figures and tables.
In describing the evaluation of results, the authors indicate that “the second step was the quantitative analysis of the geometrical parameters” (212). Did the authors instead intend to indicate that they carried out quantitative analysis, as suggested by the calculation of Pearson correlation coefficients?
Thank you for the remark, it has been corrected
The order of the Tables is confusing. In particular, why are Tables 3 and 4 introduced before Table 2 (first introduced on lines 217, 226, 263, respectively)?
Unfortunately, the figure numbering was confused, we have renumbered the figures and tables.
*“In this paper, a comparison is performed between the manual and semi-automatic segmentations performed by non-medical and medical investigators, respectively” (16-18). Through this approach, you introduce two confounding factors: (1) method of segmentation (manual vs. semi-automatic); (2) training of personnel (non-medical vs. medical). As such, your methodological approach – without further work – does not readily allow for distinction between the influence of each of these influential factors. How do you decouple these factors? The authors emphasize in the discussion that “one of the main objectives was to investigate the respective segmentation capabilities of an experienced medical and a non-medical investigator” (330-331), yet they haven’t precisely done so – this would have required the two groups of investigators to use the same method of segmentation. Elsewhere in the discussion, the authors note that “the variability due to the different segmentation methods was the aim of the investigation” (339-340), yet they haven’t precisely shown this either – this would have required the same group of investigators to use the different methods of segmentation. The method is inconsistent with the stated aims – which are themselves inconsistent – as well as the claims and conclusions.
Thank you for reflecting on this aspect of the paper. From a methodological point of view your suggestion is entirely correct and we agree that from a scientific viewpoint these factors ought to be decoupled.
However, the cooperation between medical practitioners and research engineers has its own laws. First, the method used by the medical practitioner to segment the images for the analysis for any diagnostic reasons (like VascuCap in our work) is not the same as what an engineer uses. Most of the time these software are tailored for diagnostic use; from a segmentation point of view are somewhat black boxes for the medical personnel and are not used by the engineers. On the other hand, the engineers work with another set of tools to assemble a geometry from medical images which are not and most of the time never been used by a physician. According to this reasoning the mentioned factors do not have to be decoupled since in an everyday practical setting it is not realistic for a physician to use engineering software and vice versa. The question we aimed at answering is whether the collaboration between the two fields in an everyday research setting hides some dangers. Recent studies (Berg 2018, Voß 2019) highlighted how large the difference can be in segmentations between research groups. Our study emphasizes that even though today it is not a challenge to create a geometry out of medical images for simulations, caution has to be exercised on how to construct those geometries. As such, one of the closing arguments of our study - if ever simulation replaces some of the clinical studies - was to show that coworking is much more reliable compared to individual parallel working groups in the case of simulation-based findings in clinical settings.
- RESULTS
The interchangeable use of “semi-automatic”/“manual” and “medical expert”/“non-medical expert” (e.g., 240), is somewhat confusing. See also comments above regarding confounding factors.
Our study shows that software for clinical use and for research use have the same functions (lumen segmentation) but their main focus is different. In interdisciplinary projects it is crucial that the outputs of software for clinical or research use (segmented lumen) are correlated. We have added more details to the Methods and Materials section and Figure 2 to clarify the meanings of the terms “semi-automatic”/“manual” and “medical expert”/“non-medical expert”. The following changes were made
After semi-automatic segmentations manual refinement was applied to correct the boundaries at the reconstructed surface in the lumen by two medical experts in consensus reading (ZM, ZC).
Figure 2. Steps of data preparation. a) segmentation with both techniques (top VascuCap-semi-automatic segmentation by medical experts, bottom Slicer manual segmentation by non-medical experts)
Figure 4 (257) is not effective. The authors are strongly encouraged to find a different way to convey whatever information they wish for the reader to glean from these results. This could, perhaps, take the form of a scatter plot and/or Bland-Altman plot comparing the diameters (though other effective forms are certainly possible).
Thank you for the suggestion. A Bland-Altman diagram was added to the figure (now Figure 5.) on the analysis of the average diameters. The explanation on the results of the new diagram is given in the text. Our opinion is that the original diagram should still be present in the figure as it might help the reader to see differences visually.
A minor semantic issue – the authors direct the readers to “the supplementary materials” (272), though Table A1 is instead found in the appendix (Appendix A). This led to some confusion as I searched unsuccessfully for a supplementary materials section until I stumbled across the appendix at the end of my review.
Thank you for the remark, it has been corrected.
When discussing the effects of differences of segmentation on flow analysis, the authors refer to Figure 4 (307-308), though I am quite confident that they instead meant to reference Figure 5.
Unfortunately, the figure and table numbering was wrong, we have renumbered the figures and tables.
*In reporting the effect of differences on segmentation on flow analysis, the authors indicate percent differences in maximum time-averaged wall shear stress (TAWSS), then insist that “further quantitative analysis is inapplicable here, as qualitative analysis can describe the differences well” (314-315). I strongly disagree. You earlier state that “the aim of our project is computational geometry reconstruction for blood flow simulations to make it suitable for comparison with plaque image analysis…” (14-16). If your aim is to correlate CFD results with plaque distribution, then the spatial distribution of TAWSS – not simply maximum TAWSS – is critically important, making further quantitative analysis highly applicable here. The two surfaces should be mapped to each other in order to compare values throughout the vessel, rather than just comparing the maximum of each (which may not even occur at the same location).
A further evaluation on the spatial distribution of the time-averaged wall shear stresses was carried out and is summarised in a new figure (Figure 7.) Although a proper mapping of the surfaces onto each other is not possible, the mapping of data to a parametrized rectangular space (Antiga 2004) can be achieved with VMTK. Two lines (two circumferential curves) at the same altitude from the bifurcation were analysed quantitatively. These results demonstrate how differences in segmentation can lead to different results. Further explanations are added in the text.
Results are reported for velocity profiles “at the systolic time instant” (316). Systole is typically defined as the period over which the heart is ejecting blood – how did you define or select a single “systolic time instant”? (The same applies for how the timing for the diastolic velocity profile was selected.)
Definition was added into the evaluation section.
The velocity values reported in Figure 5 (322) seem unrealistically low. Typical internal carotid artery velocities are 0.4 m/s (diastole) to 1.25 m/s (systole) in normal areas and 1-2.3 m/s (diastolic-systolic) in highly stenosed areas, yet you report values all less than ~1 m/s (and, most concerning, less than 0.5 m/s during systole in the stenosed region, P2, of Case 010). How do you explain this discrepancy?
Thank you for pointing it out. The velocities are unusually low because we had inadequate information on the boundary conditions and we used standardised ones. At the CCA the velocities agree with the measured ones. However, this problem is irrelevant since we compare simulations with identical boundary conditions and conclusions drawn from the comparisons are still valid.
- DISCUSSION & CONCLUSION
You seem to suggest that Group III yielded preferable outcomes to Group I. However, Group III had “non-significant and almost zero correlation” in minimum diameter (282; Table 3), and “moderate and strong correlations” between average diameter difference and calcification metrics were found for this group (295-296; Table 4). First, further explanation of how correlation decreased alongside relative minimum diameter difference is warranted, as this is an unexpected and peculiar observation. Second, how do you justify recommending the course of action resulting in Group III’s results when outcomes seem so poor?
We apologize, indeed the idea behind this aspect needs a better explanation, a new paragraph was added to the Discussion section. The result of Group III has been explained there as well.
“The minimum diameter is a local parameter and with increasing severity of calcification it is progressively harder to get it properly by manual segmentation. This is shown by the summarising Figure 4 and the left panel in Figure 5. (DDV curves). In Group II when we worked together, care was taken for the most problematic sections and this in turn improved the estimation of minimum diameter. Although the correlation in Group III is not significant because of the low number of available patients, our rationale is the following. Due to the collaboration in Group II, we have a good correlation for the minimum diameter as we could verify each other's work in person. This idea complements the results of the correlation analysis for the calcification metrics since when we worked together there was a significantly poor correlation between the metrics and ADDV values. This is one of the most important results of the study. The lack of correlation between calcification and average difference (ADDV) is due to the improved quality of the collaborative segmentation: none of the participants displays a bias in any direction because of the calcification.In contrast, the results for the other two groups showed a good correlation with the calcification metrics and a poor correlation for the minimum diameter, because of the inversion of the above argumentation. The main difference between the Group I and Group III is that the overall segmentation improved because of what had been learned during the collaboration phase. “
We have uploaded the revised manuscript in word file format. The changes are highlighted with red in the text. We hope you find the changes satisfactory. Unfortunately, the preparation of additional calculations and extra figures to appropriately answer needed more time as we had expected. Please, let us know if there is a need for any additional changes.
Best regards
Zsuzsanna Mihály MD

Round 2
Reviewer 2 Report
I appreciate and commend the thoughtful consideration and detailed responses to my many questions and recommendations. Changes to the manuscript, including clarifications of the method and additional figures, have made the article clearer and more compelling.
I find the responses to be mostly satisfactory, though don’t believe that some of the explanations in the responses have been adequately reflected in the manuscript. In particular, my reservation regarding the alignment of stated aims and study design have not been addressed within the manuscript. At a minimum, the presence of confounding factors – method of segmentation (manual vs. semi-automatic) and training of personnel (non-medical vs. medical) – should be noted as a limitation to guide interpretation of your study. Other claims should be reviewed for consistency and validity.
Furthermore, upon my first reading, my perception was that you were ultimately concluding that manual, non-medical expert segmentation should be performed after working together with medical experts – corresponding to the “Group III” scenario. That perception was particularly derived from the following sentence in the “Conclusion” section:
“A collective review of the medical images should precede a set of manual or automatic segmentations before using them in Computational Fluid Dynamics simulations to ensure a solid basis for proper comparison with plaque imaging analysis.”
Based upon the recent clarifications and modifications, I understand that you are instead recommending a scenario corresponding to “Group II.” I believe it may be helpful to the reader to explicitly tie your recommendation to the Group II scenario to ensure this key takeaway is clearly and unambiguously conveyed. (This could be done in the Discussion, Conclusion, or both sections.)
Author Response
Dear Reviewer2,
thank you for the minor revision of our manuscript. The manuscript reviewed text has been attached.
Our detailed answeres are below in Italic:
Thank you for revision, The Discussion and Conclusion sections were corrected to clearly interpret the questioned parts. Additional paragraph and sentences were added to the manuscript to enhance the understanding of the concerned parts.
I appreciate and commend the thoughtful consideration and detailed responses to my many questions and recommendations. Changes to the manuscript, including clarifications of the method and additional figures, have made the article clearer and more compelling.
I find the responses to be mostly satisfactory, though don’t believe that some of the explanations in the responses have been adequately reflected in the manuscript. In particular, my reservation regarding the alignment of stated aims and study design have not been addressed within the manuscript. At a minimum, the presence of confounding factors – method of segmentation (manual vs. semi-automatic) and training of personnel (non-medical vs. medical) – should be noted as a limitation to guide interpretation of your study. Other claims should be reviewed for consistency and validity.
We added a paragraph to the discussion section with the limitation of our study.
Furthermore, upon my first reading, my perception was that you were ultimately concluding that manual, non-medical expert segmentation should be performed after working together with medical experts – corresponding to the “Group III” scenario. That perception was particularly derived from the following sentence in the “Conclusion” section:
“A collective review of the medical images should precede a set of manual or automatic segmentations before using them in Computational Fluid Dynamics simulations to ensure a solid basis for proper comparison with plaque imaging analysis.”
The conclusion section of our abstract explained our views more clearly: “ A collective review of the medical images should be preceded by a set of manual segmentations before applying in computational simulations in order to ensure a proper comparison with plaque imaging analysis.”
We have refined the sentence about this statement in the “Conclusion” of the main text.
Based upon the recent clarifications and modifications, I understand that you are instead recommending a scenario corresponding to “Group II.” I believe it may be helpful to the reader to explicitly tie your recommendation to the Group II scenario to ensure this key takeaway is clearly and unambiguously conveyed. (This could be done in the Discussion, Conclusion, or both sections.)
We are sorry about not being clear enough with our main message. We have added some more explanation about it in the discussion and conclusion section.
Thank you for your revisions.
Best regards
Zsuzsanna Mihály MD
